# Development of Postoperative Ocular Hypertension After Phacoemulsification for Removal of Cataracts in Dogs

**DOI:** 10.3390/ani15030301

**Published:** 2025-01-22

**Authors:** Myeong-Gon Kang, Chung-Hui Kim, Shin-Ho Lee, Jae-Hyeon Cho

**Affiliations:** 1Hangang Animal Medical Center, Namyangjusi 12126, Republic of Korea; hangangah@hanmail.net; 2Institute of Animal Medicine, College of Veterinary Medicine, Gyeongsang National University, Jinju 52828, Republic of Korea; kimchi3237@gnu.ac.kr; 3Department of Companion Animal Health, Tongmyong University, Busan 48520, Republic of Korea

**Keywords:** dog, intraocular pressure (IOP), cataract, glaucoma, phacoemulsification (PHACO)

## Abstract

This study monitored intraocular pressure (IOP) in 31 dogs (48 eyes) after cataract surgery in 2021. Postoperative ocular hypertension (POH), defined as an IOP ≥ 25 mmHg, was observed to be higher within the first 3 h compared to later weeks. The study suggests that POH within 3 h post-surgery may indicate secondary glaucoma, highlighting the need for timely treatment.

## 1. Introduction

Phacoemulsification (PHACO), a procedure in which an ultrasonic device is utilized to fragment the opacified lens into small pieces that are subsequently aspirated from the eye, combined with the implantation of an artificial intraocular lens, represents the standard of care for cataract removal and restoration of emmetropic vision [1]. Postoperative complications can occur, including cases of elevated intraocular pressure observed as early as five hours following surgery [2]. A frustrating complication after cataract surgery is postoperative ocular hypertension (POH) [3,4]. The effects of POH on the eyes of dogs have not yet been clearly elucidated, but POH is reported to potentially damage the sensitive retina and optic nerve, resulting in reduced vision [3,5]. The causes of this vision loss include various factors such as trabecular meshwork swelling due to surgery, aqueous outflow obstruction by zonular fragments, viscoelastic agents, obstruction of the trabecular meshwork by residual lens particles, soluble lens proteins, pigment, vitreous humor, inflammatory debris, or viscous plasmoid aqueous, insufficient aspiration of viscoelastic material, use of intracameral carbachol, rapid aqueous humor production in combination with a water-tight wound closure, and narrowing of the iridocorneal angle postoperatively [6,7]. Among these factors, viscoelastic materials cause intraocular trabecular meshwork obstruction postoperatively, resulting in a decrease in outflow of aqueous humor, which leads to an increase in IOP. If the intraocular pressure increases to 25 mmHg or higher, it is defined as POH, but glaucoma treatment is usually performed only if the IOP value is 30 mmHg or higher. It was reported that as the IOP increased postoperatively, the incidence of POH was 18% five hours after surgery [8,9,10].

Miller [7] evaluated eyes histologically after phacoemulsification and found that blood had refluxed into the collecting channels and corneoscleral trabecular meshwork, the ciliary cleft cross-sectional surface area and width significantly decreased immediately after surgery, and structural alterations in the trabecular meshwork persisted even after the IOP normalized.

Chahory [10] defined POH as an IOP > 25 mmHg after cataract surgery, and Smith [11] reported that the incidence rate of POH was 48.8%. It has been shown that repeated occurrences of POH are associated with a higher risk of secondary glaucoma [11,12]. Dustin [13] argued that if three or more IOP spikes occur within 2 weeks after surgery, treatment for glaucoma should be performed, and Eva [14] diagnosed secondary glaucoma by using the occurrence of POH within 1 week after cataract surgery as a diagnostic criterion. As demonstrated by previous studies, an increase in the IOP starting from the first few hours after surgery is highly correlated with the occurrence of glaucoma and eventually causes vision loss.

This study observed and compared postoperative changes in IOP over time by measuring the IOP at 1, 2, 3, and 20 h, and at 1, 2, 3, 4, and 8 weeks postoperatively in 31 dogs (48 eyes) that underwent cataract surgery. Based on these results, the recovery rate will be improved, and complications after cataract surgery in dogs will be reduced by identifying the optimal timing for diagnosing secondary glaucoma in relation to the onset of POH.

## 2. Materials and Methods

This study is a retrospective analysis on postoperative changes in IOP in 31 dogs (48 eyes) that underwent cataract surgery after visiting the Hangang Animal Medical Center in Namyangju City, Gyeonggi Province during a one-year period. The breed, sex, age, and body weight of the 71 of dogs that underwent surgery are reported in Table 1. In the 2022–2023 study, the characteristics of the 31 dogs (48 eyes) that underwent cataract surgery are presented. Nine of the participating dogs were diagnosed with diabetes, while the remaining dogs were healthy and showed no signs of uveitis or other ophthalmic conditions apart from cataracts, as noted in Table 1. Dogs that underwent cataract surgery for reasons such as the correction of anterior or posterior lens luxation, lens extraction due to lens damage, or preventive cataract surgery on a blind eye to avert lens-induced uveitis were excluded.

Several tests were performed prior to cataract surgery, and the severity of the cataracts in the dogs was examined by slit-lamp biomicroscopy (Topcon, Tokyo, Japan) to classify the stages of the cataracts in the dogs. As a result, the cataract stages in the dogs were classified into four grades: Incipient, Immature, Mature, and Hypermature (Table 2). The IOP was measured by tonometry (Tonovet, Finland), and surgery was performed when the IOP value did not exceed 25 mmHg. Exceptionally, the 15th subject underwent surgery despite a high IOP of 41 mmHg, as its IOP returned to a normal level within 2 h without the administration of any medication (Table 3).

A fundus examination was performed with an indirect ophthalmoscope (Keeler, Malvern, PA, USA), and an electoretinogram (ERG) test was conducted with ERG equipment (RETI port 3s ERG, an-vision, Wiesbaden, Germany). In addition, the retinal detachment status was examined using ultrasonography (ARIETTA65, Fujifilm, Tokyo, Japan). The examination results showed that all the subjects showed normal results.

For both diabetic and nondiabetic patients, Eyelebo (Levofloxacin 0.5%, Hanlim Pharm., Yonginsi, Republic of Korea), Eyemetrone (Fluorometholone 0.1%, Hanlim Pharm., Yonginsi, Republic of Korea), and Optanac Eye Drops 0.1% (Diclofenac Sodium 1 mg/mL, Samil Pharm., Seoul, Republic of Korea) were administered three times daily for seven days as topical medications. To achieve sufficient mydriasis prior to surgery, Mydrin-P (0.5% Tropicamide and 0.5% Phenylephrine, Hankugsanten Pharm., Seoul, Republic of Korea) and Isopto-Atropine (Atropine 1%, Alcon Pharm., Fort Worth, TX, USA) were administered topically three times at 30-min intervals, starting 2 h before surgery. For systemic administration, nondiabetic patients received Amocra^®^ (Amoxicillin Hydrate and Potassium Clavulanate 375 mg, KUHNIL Pharm., Seoul, Republic of Korea) at a dosage of 10 mg/kg BID and Prednisone (Solondo 5 mg, YUHAN Pharm., Seoul, Republic of Korea) at 0.5 mg/kg BID for 7 days. In contrast, diabetic patients followed a steroid-free regimen, receiving Amocra at 10 mg/kg BID along with Meloxicam Chewable Tablets (Meloxicam 1 mg, Ashish Life Science Pvt Limited, Maharashtra, India) at a dosage of 0.1 mg/kg SID for 7 days. The anesthesia equipment used for surgery was the Drager Atlan A300 (Lübeck, Germany) and forced circulation was induced by the equipment for both inhalation and exhalation. When there was a need to perform bilateral cataract surgery, surgery was performed when it was judged that the patient had no problem with the operation, and surgery was performed starting with the eye with a lower stage of cataract. All operations were performed by the same veterinarian.

The same anesthesia protocol was applied to all the patients. The premedication agents used before induction of anesthesia were as follows: Atropine (atropine sulfate 0.5 mg/kg, JEIL Pharm., Seoul, Republic of Korea) 0.03 mg/kg SC, Diazepham (Diazepham inj 10 mg/2 mL, SAMJIN Pharm., Seoul, Republic of Korea) 0.5 mg/kg IV, Butopanol (Butopanol 1 mg/1 mL, Myeongmun Pharm., Seoul, Republic of Korea) 0.1 mg/kg IV, Propofol (Probio 1% Myeongmun Pharm., Seoul, Republic of Korea) 6 mg/kg IV, and Cepalzolin (cephazolin natrium 1 g) 0.05 g/kg SC. Isoflurane anesthetic gas was used to maintain anesthesia during surgery. In addition, Ringer’s Lactate solution (HARTMANN, Jungweon pharm., Seoul, Republic of Korea) was used to maintain adequate perfusion during surgery. To prevent the eyeball from rotating and maintain pupillary dilation during surgery, Acrium (Atracrium besylate, Myeongmun Pharm., Seoul, Republic of Korea), a muscle relaxant, was administered intravenously at a dose of 0.2 g/kg, and the patient was placed in a supine position.

Cataract surgery was performed using the PHACO (BAUSCH + LOMB, Stellaris Elite^TM^, St. Louis, MO, USA) method. A 2.8 mm corneal incision was made 2 mm away from the corneal limbus (Figure 1A). Before performing capsulorhexis, a viscoelastic agent (Kukje Hyaluronate-I inj., 0.85 mL/dog, Kukje, Pharm., Seoul, Republic of Korea) was injected into the intracamera to maintain the shape of the anterior chamber for the capsulorhexis and to protect the corneal endothelium. Trypan blue was utilized as an adjunct agent to enhance visualization of the anterior capsule during PHACO of mature white cataracts (Figure 1B). An incision was made in a round shape with a diameter of 5 mm in the center of the lens anterior capsule (Figure 1C). Then, the lens was separated by hydrodissection with a BSS solution (Alcon BSS^TM^, Fort Worth, TX, USA) (Figure 1D). The residues of the lens nucleus and lens cortex were aspirated by PHACO, and the lens was polished clean (Figure 1E). Intraocular lenses (an-vision, MD4-12, D+41.0, West Jordan, UT, USA) matching the size of each eye were implanted into the lens capsule using an injector. The corneal incision was closed by simple interrupted sutures with 10-0 black nylon (REXLON-black nylon USP10-0, spatula 6 mm 3/8, Seoul, Republic of Korea) (Figure 1F). During the final stage of surgery, when physiological pressure in the anterior chamber was needed, a BSS solution was injected to inflate the anterior chamber. At this time, IOP was estimated by digital palpation instead of measuring by a tonometer.

Postoperatively, Amocra, an anti-inflammatory drug, was administered BID for 7 days. For topical medications, Eyelebo 0.5% (Levofloxacin Hydrate 5 mg/mL, Genu Pharm., Seoul, Republic of Korea), Eyemetholone Eye Drops 0.1% (Fluorometholone 1 mg/mL, Genu Pharm., Seoul, Republic of Korea), and Optanac Eye Drops 0.1% were administered for up to 6 months while checking the condition of each patient.

For diabetic cataract patients, topical medications were administered according to the same protocol used in diabetic patients, and Amocra and Meloxicam Chewable Tablets, a non-steroidal anti-inflammatory drug, were also administered orally for 7 days. Postoperatively, IOP measurements and ophthalmologic examinations were carried out by tonometry and ophthalmoscopy, respectively, at 1, 2, 3, and 20 h, and at 1, 2, 3, 4, and 8 weeks after surgery. In addition, an eye examination was performed by slit-lamp biomicroscopy 1 month after surgery. After cataract surgery, Cosopt^®^ solution (Dorzolamide 2%/Timolol 0.5%, Tampere, Finland) was administered prophylactically to most patients beginning on the day of surgery and was continued for a period of three weeks to prevent postoperative hypertension (POH). Dogs with ocular hypotension were excluded from this treatment. In some cases, the dosing schedule for the Cosopt solution was adjusted throughout the three-week period according to intraocular pressure (IOP) measurements.

When the IOP did not respond adequately to Cosopt^®^ solution or there was an acute increase in the IOP, Xalatan (latanoprost ophthalmic solution 0.005%, Pfizer Inc., New York, NY, USA) was administered topically to the eye, and when the IOP did not respond within a half hour, 1 g/kg of mannitol was administered for 20 min by the intravenous bolus injection method. When the IOP did not decrease even after urination, paracentesis was used in combination with medication. Also, when there was an increase in the IOP due to intraocular fibrin formation, paracentesis was performed, and 25~50 mg of TPA was injected into the eye through the limbus using a 30-gauge needle. In addition, when there was a risk of adhesion in the iris postoperatively, Mydrin-P, a topical mydriatic drug, was administered following surgery.

When the patient had eyesight even with an increased IOP, an Ahmed valve was additionally installed, and cataract surgery was performed using the A.R.C laser system (Diode laser FOX). The A.R.C. laser system was used with reference to the literature as follows. The average parameter is 2 watts at 2 s, and an average of 12~18 coagulation spots may be applied. The hand piece needs to be placed properly about 1 mm from the limbus with light pressure to aim and target the cells. The 3 and 9 o`clock positions should be left blank to protect the ciliary nerves. Cyclophotocoagulation was used for glaucoma surgery, and the laser system had to be equipped with the handpiece (HS11025s). The dog owners were advised to have their dogs to wear an Elizabethan collar for 3 weeks to restrict the activities of dogs that underwent surgery.

## 3. Results

In Table 2, data are provided on the preoperative conditions of each eye of the subjects. The mean age of the subjects was 7.9 ± 0.52 years and the mean body weight was 5.54 ± 0.4 kg. When performing PHACO, it was not possible to separately measure PHACO time for the oculus dexter (OD, Right eye) and oculus sinister (OS, left eye) during the process of lens extraction since operations on the OD and OS were performed continuously, so PHACO time was measured for each dog by using one dog as the unit from the data in the study from 2022–2023. The mean PHACO time was 3.81 ± 0.42 min.

In 31 dogs (48 eyes) that underwent surgery, intraocular pressure (IOP) was measured at 1, 2, 3, and 20 h, as well as at 1, 2, 3, 4, and 8 weeks post-surgery. The data were categorized into three groups: Group A, B, and C. Group A included pre-surgery data. Group B included post-surgery data at 1, 2, and 3 h. Group C included post-surgery data from 20 h to 8 weeks. The mean IOPs for Group A, B, and C were 13.10 ± 8.29 mmHg, 17.84 ± 5.33 mmHg, and 13.71 ± 4.78 mmHg, respectively, as shown in Figure 2.

Significant differences in IOPs were observed among the groups (*p* < 0.001). Group B demonstrated significantly higher IOPs compared to both Group A and Group C (*p* < 0.01 and *p* < 0.001, respectively). No significant difference was observed between Group A and Group C. (Figure 2).

Paired tests were conducted to compare pre-surgery IOPs with each post-surgery IOP measurement. The median IOP values for the pre-surgery and post-surgery time points (1 h to 8 weeks) were as follows: 15.23 ± 11.32, 17.45 ± 12.42, 15.61 ± 7.71, 11.81 ± 9.03, 8.48 ± 7.01, 11.39 ± 12.39, 10.12 ± 8.73, 9.58 ± 7.90, 9.48 ± 8.15, and 9.45 ± 8.02. Each significance was identified for the pre-surgery IOP measurements and for the IOP measurements at 1 (*p* < 0.05), 2 (*p* < 0.01), and 3 (*p* < 0.01) weeks post-surgery in Figure 3. However, 8 weeks after surgery, total vision loss occurred in five dogs (six eyes). The cataract surgery had good outcomes in 26 dogs (42 eyes) out of the total 31 dogs (48 eyes), so the surgery success rate was 83.88% (87.5%).

## 4. Discussion

This study, conducted from 2022 to 2023, aimed to evaluate the risk of vision loss in dogs following cataract surgery by monitoring the onset and progression of POH to glaucoma. By identifying the timing of POH and secondary glaucoma, this study sought to reduce vision loss through prompt medication and prevent IOP spikes with prophylactic treatment to maintain stable intraocular pressure (IOP).

In general, the normal IOP level in dogs is reported to range between 12 and 25 mmHg. However, cataract surgery is known to cause IOP elevation from 3 to 5 h postoperatively [3,10,15]. Postoperative ocular hypertension (POH) is defined as a postoperative IOP of 25 mmHg or higher [3]. This definition was applied in this study as it aligns with the experimental objectives.

Previously identified risk factors for POH included specific breeds, prolonged surgical duration, advanced age, intraocular lens implantation, hypermature cataract stage, pre- and postoperative inflammation, uveal or retinal abnormalities, intraocular hemorrhage, intracameral carbachol use, and certain topical anti-inflammatory medications [7,10,16,17,18].

Dustin [12] defined POH as an IOP exceeding 20 or 25 mmHg and divided subjects into treatment groups receiving latanoprost, dorzolamide, or dorzolamide/timolol postoperatively. Dustin’s study reported no significant differences in POH reductions among the treatment groups compared to controls. However, it was noted that the dorzolamide/timolol combination reduced the incidence of POH when administered three times—at 14 and 2 h preoperatively and at corneal incision closure [9]. In this study, even with aggressive postoperative use of dorzolamide/timolol, IOP elevations were observed, necessitating the use of mannitol in some cases.

Interestingly, the use of intracameral tissue plasminogen activator significantly reduced the development of POH at levels exceeding 25 mmHg. In human medicine, intracameral tissue plasminogen activator has long been used to address fibrin formation in the anterior chamber following cataract surgery. In veterinary medicine, its application remains empirical, focusing on the prevention or treatment of fibrin formation post-canine phacoemulsification [19].

Administration of tissue plasminogen activator into the anterior chamber immediately after surgery was shown to reduce fibrin formation in dogs undergoing phacoemulsification cataract surgery, thereby improving surgical outcomes [19]. Fibrin formation after cataract surgery is associated with an increased POH due to posterior synechiation, peripheral anterior synechiation, and/or obstruction of aqueous humor drainage pathways. In this study, 25–50 mg of tissue plasminogen activator was administered over 14 days postoperatively to mitigate fibrin formation, which effectively reduced both fibrin and IOP levels.

Dustin [13] regarded three or more IOP spikes within two weeks after cataract surgery as a diagnostic criterion for glaucoma, while Eva [14] diagnosed glaucoma based on an IOP of 25 mmHg or higher within seven days post-surgery. These findings from previous studies aligned with the results of the present study. As shown in Figure 2, an early response to IOP within 3 h (Group B) was found to be more critical than the response after 20 h (Group C). Figure 2 illustrates the importance of early intervention in cases of elevated IOP. In individual case evaluations, 2 out of 48 eyes were diagnosed with POH three or more times within two weeks, and two additional cases were diagnosed with POH within one week. These cases were identified as requiring follow-up and further treatment.

Postoperative measurements revealed elevated IOP levels at and around three hours post-surgery. This finding corroborates data from Manuela [9], who reported that POH peaked two hours after cataract surgery and subsequently decreased. In this study, while the IOP was significantly elevated compared to preoperative levels at three hours, it was reduced substantially by 20 h (11.77 ± 7.55, *p* < 0.058).

Berson [20] reported that the use of sodium hyaluronate in the anterior chamber reduced aqueous humor production by approximately 65%. Chahory [3] noted that, among recently developed viscoelastic materials, sodium hyaluronate caused the least increase in IOP, with a POH incidence of 18%. The IOP significantly decreased one-hour post-surgery but rose again between three and five hours. Based on these findings, sodium hyaluronate was employed in this study to minimize POH. When the postoperative IOP exceeded 25 mmHg, topical Cosopt^®^ (dorzolamide hydrochloride 2.0%/timolol maleate 0.5%) was administered, effectively reducing the IOP in most cases. However, dogs were discharged three hours post-surgery, and owners were instructed to apply Cosopt^®^ twice daily. As a result, IOP measurements beyond three hours were not feasible, limiting direct comparisons with other studies. Unlike prior findings, Figure 3 shows a significant increase in IOP over the 3 h period post-surgery.

The success rate of cataract surgery in diabetic dogs has been reported to be comparable to that in non-diabetic dogs [21,22]. However, spontaneous lens capsule rupture and subsequent phacoclastic uveitis are more prevalent in diabetic dogs, potentially leading to phacomorphic glaucoma due to an intumescent lens and shallow anterior chamber [22,23].

In this study, nine dogs (18 eyes) with diabetes mellitus were included, among which vision loss occurred in three dogs (four eyes), reflecting a vision loss rate of 22.22%. This rate was approximately 10–14% higher than the overall vision loss rate of 12.5% (six eyes from 31 dogs). These findings also highlighted a higher incidence of postoperative glaucoma in diabetic dogs compared to their non-diabetic counterparts.

The overall success rate of cataract surgery in 31 dogs (48 eyes) was 87.5%. Although the sample size was small, this success rate aligned with the 84.5% rate reported by Harathi [24].

## 5. Conclusions

Good clinical outcomes after cataract surgery in small animals require intensive management of intraocular pressure (IOP) elevations within the first 24 h post-surgery. Regular follow-up check-ups should be conducted for at least 8 weeks after surgery. In cataract surgery, postoperative hypertension (POH) may occur within 1 to 3 h post-surgery due to an increase in IOP caused by various factors, such as residual lens particles, soluble lens proteins, inflammatory debris, lens-induced uveitis, and viscoelastic materials. Therefore, if an elevated IOP is observed within the first day post-surgery, it should be diagnosed as secondary glaucoma and treated prophylactically to improve recovery outcomes following cataract surgery.

## 6. Statistical Analysis

This study performed statistical tests randomly due to intra-case correlation and presented the statistical values. For each test, *p* < 0.05 was considered statistically significant. After cataract surgery by PHACO, statistical analysis was performed using a paired *t*-test to evaluate the changes over time from Pre-OP to Post-4W following the surgery. Additionally, intraocular pressure (IOP) was compared among three groups: pre-surgery (Group A), 1 to 3 h post-surgery (Group B), and 20 h to 8 weeks post-surgery (Group C). The non-parametric Friedman test was used to evaluate overall differences between the groups, followed by pairwise comparisons using the Wilcoxon signed-rank test.

## Figures and Tables

**Figure 1 animals-15-00301-f001:**
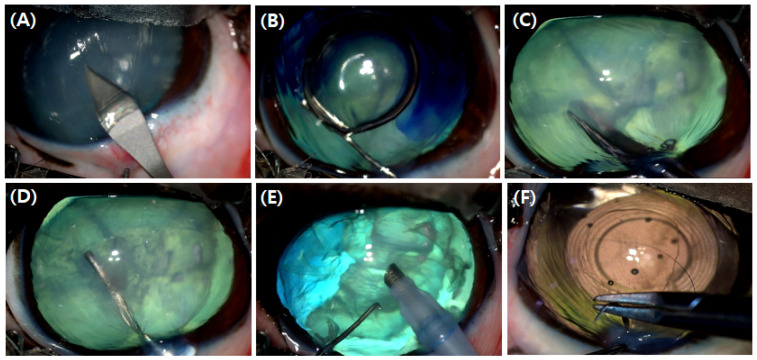
PHACO for cataract removal: (**A**) corneal incision, (**B**) staining of anterior lens capsule using trypan blue, (**C**) round incision with diameter of 5 mm in the center of the eye lens anterior chamber, (**D**) hydrodissection to separate the eye lens, (**E**) removal of the lens nucleus using phaco handpieces, and (**F**) intraocular lens implantation and corneal suture.

**Figure 2 animals-15-00301-f002:**
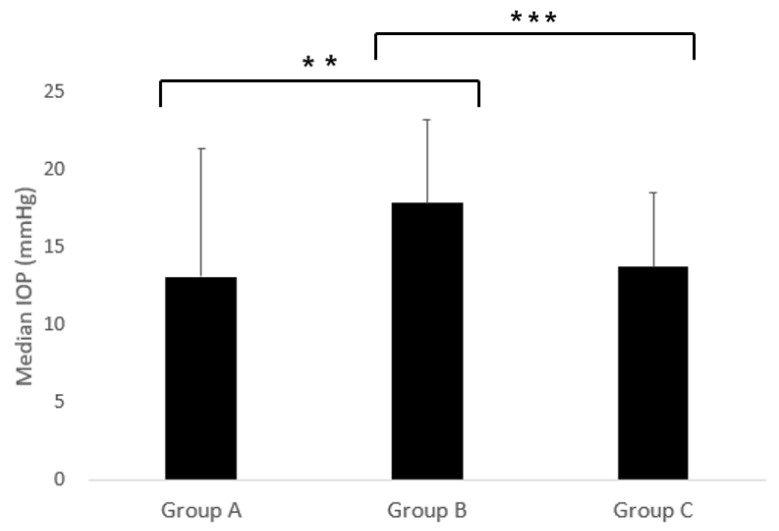
The intraocular pressures (IOPs) of the dogs that underwent surgery were measured according to the follow-up periods (Group A, B, and C). Values are presented as mean ± standard deviation (SDs). The *p*-values were obtained using the non-parametric Friedman test, followed by pairwise comparisons with the Wilcoxon signed-rank test. *p* < 0.05. Group A: preoperative data; Group B: data from 1 to 3 h post-surgery; and Group C: data from 20 h to 8 weeks post-surgery. ** *p* < 0.01, *** *p* < 0.001.

**Figure 3 animals-15-00301-f003:**
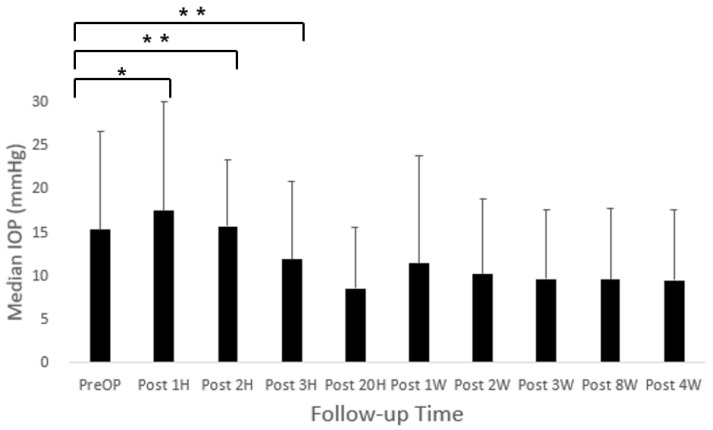
Time course of intraocular pressure (IOP) after cataract surgery using PHACO in dogs. Data are expressed as mean ± SD, *p*-value was acquired by paired *t* test. * *p* < 0.05, ** *p* < 0.01. Pre-OP: Pre-surgery, Post: Post-surgery.

**Table 1 animals-15-00301-t001:** Type of cataract of the 31 dogs.

No	Breed	Sex	Age (Year)	BW (kg)	History
1	Yorkshire Terrier	M	10	6	DM
2	Yorkshire Terrier	F	11	2.4	
3	Shih Tzu	M	11	7	DM
4	Poodle	M	7	9.4	
5	Poodle	M	5	4.6	
6	Poodle	M	6	5.7	
7	Poodle	M	13	5	
8	Mongrel	M	12	4	
9	Mongrel	F	9	9.8	DM
10	Maltese	M	6	4.4	
11	Poodle	F	9	4.1	
12	Poodle	M	7	5.4	
13	Poodle	F	12	4	
14	Dachshund	F	5	11	DM
15	Yorkshire Terrier	F	11	2.4	
16	Maltese	F	5	6.6	
17	Pomeranian	M	6	3	
18	Poodle	F	5	6.7	
19	Maltese	F	11	4.1	DM
20	Maltese	M	11	2.2	DM
21	Maltese	M	6	5.8	
22	Bichon Frise	M	5	7.1	
23	Maltese	M	7	6	
24	Cocker	M	6	8.5	
25	Poodle	M	6	2.6	DM
26	Maltese	M	12	4.4	
27	Maltese	F	11	3.6	
28	Chihuahua	M	7	6.3	DM
29	Poodle	M	5	6.9	DM
30	Maltese	F	5	6.6	
31	Bichon Frise	F	3	6.2	

BW: Body weight. DM: Diabetes mellitus.

**Table 2 animals-15-00301-t002:** Ocular examination and data on the operated eyes for the 31 dogs.

No	Ocular	Stage of Cataract		Operated		Time PHACO	Postoperative
	Examination	RE	LE	Eye		(Minutes)	Complications
1	bilateral cataract	mature	mature	OD	OS	3.31	posterior synechiae (OU)
2	bilateral cataract	immature	mature		OS	5.5	
3	bilateral cataract	mature	hypermature	OD	OS	7.5	Glaucoma (OD)
4	bilateral cataract	mature	immature	OD	OS	5.04	Anterior chamber fibrosis (OD)
5	bilateral cataract	immature	mature	OD	OS	0.96	PC tear (OD)
6	bilateral cataract	-	mature		OS	1.45	
7	bilateral cataract	mature	hypermature	OD		3.13	PC tear (OD), vitrectomy
8	bilateral cataract	mature	mature	OD	OS	7.47	
9	bilateral cataract	mature	mature	OD	OS	1.54	corneal ulcer (OU)
10	bilateral cataract	hypermature	mature	OD	OS	2.16	
11	unilateral cataract	mature	-	OD		3.58	Intraocular hemorrhage, adhesion in iris (OD)
12	bilateral cataract	immature	mature	OD	OS	1.21	
13	bilateral cataract	mature	hypermature	OD	OS	6.17	Retinal detachment
14	bilateral cataract	mature	mature	OD	OS	1.13	
15	unilateral cataract	mature	-	OD		6.11	
16	bilateral cataract	mature	immature	OD		0.78	
17	unilateral cataract	mature	-	OD		4.35	
18	bilateral cataract	hypermature	mature	OD	OS	3.46	LZ rupture, anterior hemorrhage (OS)
19	bilateral cataract	mature	mature	OD	OS	2.44	
20	bilateral cataract	mature	mature	OD	OS	6.16	Glaucoma (OS)
21	unilateral cataract	-	mature		OS	4.2	
22	bilateral cataract	-	immature		OS	0.96	
23	bilateral cataract	hypermature	mature	OD	OS	5.49	
24	unilateral cataract	mature	-	OD		2.16	AC tear
25	bilateral cataract	mature	mature	OD	OS	3.15	Glaucoma (OU)
26	unilateral cataract	mature	-	OD		2.16	Iridocyclitis
27	bilateral cataract	mature	incipient	OD		9.33	Vitreous body blood
28	bilateral cataract	immature	mature	OD	OS	3.39	
29	bilateral cataract	immature	mature	OD	OS	1.44	
30	unilateral cataract	-	mature		OS	1.31	
31	unilateral cataract	immature	-	OD		2.43	

PHACO: Phacoemulsification, OD: Oculus dexter (right eye), OS: Oculus sinister (left eye), OU: Oculus uterque (a pair of eyes), AC: Anterior capsule, PC tear: Posterior capsule tear, LZ lupture: Lens Zonules rupture.

**Table 3 animals-15-00301-t003:** Changes in intraocular pressure (IOP) following cataract surgery.

No	Pre-OP		Post 1 H		Post 2 H		Post 3 H		Post 20 H		Post 1 W		Post 2 W		Post 3 W		Post 4 W		Post 8 W	
	OD	OS	OD	OS	OD	OS	OD	OS	OD	OS	OD	OS	OD	OS	OD	OS	OD	OS	OD	OS
1	12	9	10	5	17	16	17	15	11	10	12	11	15	16	14	13	14	15	14	15
2	-	14	-	19	-	17	-	13	-	13	-	20	-	14	-	14	-	18	-	13
3	14	12	29	25	17	20	16	15	9	13	13	13	17	11	12	10	13	11	35	11
4	18	16	35	37	24	37	24	6	7	24	11	11	14	14	12	9	13	14	14	11
5	15	12	15	51	15	20	15	27	15	51	13	13	15	17	13	13	15	15	19	16
6	-	15	-	23	-	17	-	21	-	15	-	30	-	13	-	12	-	20	-	12
7	15	-	21	-	19	-	10	-	7	-	8	-	7	-	6	-	13	-	10	-
8	13	11	12	16	16	19	20	21	7	12	9	13	11	11	8	9	11	10	8	8
9	11	9	12	11	9	9	10	11	8	7	5	7	12	8	9	6	9	8	6	7
10	20	15	35	18	24	24	23	19	12	18	15	16	16	17	18	16	15	14	24	20
11	17	-	24	-	17	-	20	-	19	-	9	-	14	-	15	-	10	-	7	-
12	17	14	24	23	21	23	20	18	13	13	15	17	16	14	12	11	15	16	-	-
13	11	8	5	6	7	8	10	9	7	4	16	14	14	14	12	14	13	17	10	15
14	8	5	6	6	8	10	13	14	5	4	10	5	10	9	12	10	11	10	8	10
15	41	-	11	-	12	-	12	-	3	-	15	-	4	-	8	-	8	-	10	-
16	17	-	35	-	25	-	25	-	14	-	15	-	17	-	7	-	14	-	12	-
17	23	-	6	-	17	-	20	-	3	-	19	-	22	-	15	-	19	-	21	-
18	11	19	8	22	20	18	23	18	15	18	16	19	11	12	10	10	16	14	7	7
19	16	18	3	7	19	22	19	21	5	6	12	14	21	17	22	22	16	18	15	16
20	11	13	16	14	13	16	20	18	8	10	11	12	15	50	14	60	13	42	14	32
21	-	14	-	37	-	17	-	23	-	14	-	10	-	11	-	10	-	11	-	13
22	-	9	-	7	-	13	-	15	-	11	-	9	-	14	-	12	-	11	-	11
23	23	19	20	18	23	18	14	15	17	14	20	20	17	17	16	14	16	15	15	15
24	13	-	23	-	22	-	19	-	13	-	17	-	12	-	17	-	18	-	13	-
25	10	6	14	6	20	16	14	14	11	8	65	15	50	15	5	6	11	11	56	31
26	15	-	13	-	15	-	18	-	7	-	12	-	11	-	7	-	14	-	14	-
27	10	-	11	-	20	-	18	-	4	-	17	-	23	-	10	-	14	-	9	-
28	12	12	20	15	24	21	23	19	9	11	45	17	45	11	20	40	7	7	7	8
29	18	19	23	24	14	15	14	12	11	12	8	9	9	8	10	1	7	8	11	12
30	-	16	-	16	-	27	-	25	-	22	-	13	-	18	-	7	-	16	-	13
31	15	-	17	-	22	-	24	-	15	-	9	-	15	-	15	-	13	-	9	-

NO: Number, Pre-OP: Pre-surgery, Post: Post-surgery. OD: Oculus dexter (right eye), OS: Oculus sinister (left eye), H = Hour, W = Week.

## Data Availability

All data generated or analyzed during this study are included in this article.

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
