# Peer review of "Development of Postoperative Ocular Hypertension After Phacoemulsification for Removal of Cataracts in Dogs"

_animals, 2025, doi:10.3390/ani15030301_

Round 1
Reviewer 1 Report
Comments and Suggestions for Authors
The manuscript addresses a well-known topic: postoperative ocular hypertension (POH) in dogs following cataract surgery. Although the data presented are valuable, the novelty is moderate. Findings on the management of POH and risk factors for the development of glaucoma have been widely discussed in the literature. The study contributes with detailed IOP monitoring at various time intervals, but does not add significant advances or changes to the current understanding of the subject.
The experimental design is well structured with consistent IOP measurements at key points. The sample size (31 dogs) is limited, which affects the generalisability of the results. The lack of a control group also weakens the ability to draw robust conclusions about the efficacy of treatments used to manage POH.
The topic is relevant to veterinary ophthalmology.Nevertheless the clinical impact is limited because the results confirm findings that are already well documented. The use of tissue plasminogen activator (TPA) to prevent fibrin formation is an interesting aspect that could be emphasised more, as this practice is less common in veterinary practice.
The statistical methods are appropriate, although a more critical discussion of the results in terms of clinical relevance would be beneficial.
Comments on the Quality of English Language
The English in the manuscript is generally understandable but requires revision to improve fluency, clarity, and precision. There are multiple instances of awkward phrasing, incorrect verb tenses, missing articles, and sentence structures that hinder the flow of ideas. Additionally, certain scientific terms and methods are presented in a way that could confuse the reader.
Some examples:
L13. This study observed postoperative changes in intraocular pressure (IOP) after performing lens extraction by phacoemulsification and the implantation of intraocular lens (IOL) in 31 dogs (48 eyes) with cataracts that visited a veterinary hospital.
L22. IOP value in pre-surgery was higher in POH cases occurring from post-surgery 0 to 20 hours after surgery than in post-surgery 1 week or more after surgery (p<0.05).
L62. Based on these results, we intend to improve the recovery rate and reduce the complication after cataract surgery in dogs by suggesting the time to diagnose secondary glaucoma in relation to the occurrence of POH.
L84. However, exceptionally, the 15th subject received surgery despite a high IOP value of 41 mmHg, because its IOP returned to a normal level after 2 hours without administration of any medication.
L139. Trypan blue has been used as an adjunct for improving visualization of the anterior capsule during PHACO of mature white cataracts.
L234. It is defined postoperative ocular hypertension (POH) as a postoperative IOP of 25 mmHg or higher.
L287. Unlike previous studies, as shown in figure 3, significant IOP elevation was seen up to 1-3 hours after surgery.
In several parts of the manuscript, articles like "the" and "a" are either missing or misused. A review focusing on the proper use of articles would improve the clarity of the text.
The manuscript switches between abbreviations and full terms inconsistently. For example, "PHACO" should either be used consistently throughout or the full term "phacoemulsification" should be used.
Author Response
Dear Dr.
Thank you very much for your review. I am attaching it for your reply. We have tried to make the sentences cleaner with your advice. We've also made changes to the text and reviewed your use of a and the. We have also unified them all with the abbreviation PHACO.
Best regard.

Reviewer 2 Report
Comments and Suggestions for Authors
Dear authors, thank you for submitting this interesting manuscript. Your study, which aims to monitor and compare postoperative IOP changes over time in dogs undergoing cataract surgery, is well structured and detailed. In my opinion, it could make an important contribution to improving the recovery rate and reducing complications after cataract surgery in dogs. The clarity of presentation makes the manuscript a stimulating and useful read for both researchers and clinicians in the field. Congratulations on your work.
I just have a few minor observations:
Line 30: “In the surgical treatment of cataracts in dogs, PHACO” Please add the explanation “phacoemulsification (PHACO)”
Lines 30-32: “In the surgical treatment of cataracts in dogs, PHACO with implantation of an artificial intraocular lens (IOL) is the standard of care for removal of cataracts and restoration of emmetropic vision” Please add bibliography
Lines 36-37: “The causes of such vision loss are reported to include various factors“ I think it may sound better as follows “The causes of this vision loss include various factors”
Lines 45-47: “If intraocular pressure is increased to 25 mmHg or higher, it is defined as POH, glaucoma treatment is performed if the IOP value is 30 mmHg or higher, and it has been reported that as IOP increased postoperatively, the incidence of POH was 18% 5 hours after surgery [7, 8, 9].” I think it may sound better as follows “If intraocular pressure is increased to 25 mmHg or higher, it is defined as POH but glaucoma treatment is usually performed only if the IOP value is 30 mmHg or higher. It has been reported that as IOP increased postoperatively, the incidence of POH was 18% 5 hours after surgery [7, 8, 9].”
Lines 71-72: “The breed, sex, age and body weight 71 of dogs that underwent surgery are presented” I think it may sound better as follows “The breed, sex, age and body weight 71 of dogs that underwent surgery are reported in Table 1”
Lines 72-76: “Although 9 dogs had diabetes, other dogs were healthy, and they were free of uveitis or other ophthalmic diseases than cataract in table Dogs that underwent cataract surgery for correction of anterior or posterior lens luxation, lens extraction due to lens damage, or cataract surgery on the blinded eye to prevent lens-induced uveitis (LIU) were excluded from this case study.” Please rewrite this period
Lines 96-97: “The examination results showed that all the subjects 96 showed normal fundus images and had a normal ERG” Please avoid repetitions
Lines 100-114: Therapy should be listed neatly, indicating dosage and timing. The authors could group topically administered drugs first. Afterwards, those administered orally or systemically. The administration of Amocra, for example, is repeated several times (first at line 103 and then again at line 108). Please rewrite and clarify this period
Lines 107-108: “However, since steroidal agents cannot be administered to diabetic cataract patients, systemic Amocra was administered at a dose of 10 mg/kg BID”
Lines 127-128: “Ringer Lactate solution (HARTNANN, JUNGWEON, PHA. South Korea was used to maintain adequate […]” Please close the open parenthesis
Lines 150-151: “Postoperatively, Amocra, an anti-inflammatory drug, was administered BID for 7 days.” At lines 103-104 the authors describe Amocra as an antibiotic, not anti-inflammatory drug ‘In addition, systemic Amocra (Amoxicillin Hydrate & potassium clavulanate 375 mg, KUHNIL Pharm., South Korea)’
Lines 166-170: “After cataract surgery, CosoptⓇ solution (Dorzolamide 2%/timolol 0.5%, Tampere, Finland) was used prophylactically for about 3 weeks from the day ofsurgery in most patients except for dogs with ocular hypotension to prevent POH, and in some patients, the administration of Cosopt solution was adjusted depending on the IOP level.”. Please rewrite and clarify this period
Line 172: “Xalatan (Xalatantm” If ‘tm’ stands for ‘trade market’ add the appropriate symbol
Lines 191-193: “The characteristics of 31 dogs (48 eyes) that underwent cataract surgery are presented in table In table 2, data were provided data on the preoperative conditions of each eye of subjects.” Please avoid repetitions, rewrite and clarify the period
Lines 195-196: “The mean age of the subjects was 7.9 ± 0.52 years, and the mean body weight was 5.54 ± 0.4 kg. When performing PHACO, it was not possible to separately measure PHACO time for OD and OS during the process of lens extraction since operations on OD and OS were performed continuously, so PHACO time was measured for each dog using 1 dog as the unit.” What is meant by ‘OD’ and ‘OS’? Please explain the acronyms, rewrite and clarify the period
Lines 227-231: “This study aimed to evaluate the probability of vision loss due to the occurrence of POH and the progression of POH to glaucoma after cataract surgery in dogs to identify the time of development of POH and secondary glaucoma and minimize vision loss by rapid administration of medication. This study also intended to minimize vision loss by preventing IOP spikes through prophylactic administration of medication to adjust IOP.” Please avoid repetitions, rewrite and clarify the period
Lines 235-236: “Since this definition is relevant to the purpose of this study, it was referred to regarding the experimental method.” I don't think it's necessary to specify this
Lines 250-251: “Interestingly, though, the use of intracameral tissue plasminogen activator significantly decreased development of POH 25 mmHg” Please rewrite and clarify the period
Lines 251-254: “In humans, intracameral tissue plasminogen activator (TPA) has been utilized for many years to treat anterior chamber fibrin formation after cataract surgery. In veterinary medicine, its use is empirical either to treat or prevent fibrin formation after canine phacoemulsification [19].” Bibliographic reference 19 appears to be inconsistent with the text. If the bibliographic reference refers to the use of intracameral tissue plasminogen activator (TPA) in humans, please move this reference. For the use of tissue plasminogen activator (TPA) in veterinary medicine, please add an appropriate bibliographic reference to justify its use.
Lines 280-282: “POH. and when […]” Please delete dot

Author Response
Thank you for your review. The authors have diligently responded to your review and revised the study accordingly.

Reviewer 3 Report
Comments and Suggestions for Authors
The authors present the data as a "case presentation" when in fact it is a study of 31 dogs with cataracts. Consequently there manuscript is missing process/procedural information, and the statistics section has been relocated to the end of the paper by itself. I appears as though each eye was treated as an independent sample. That is in error. Eyes on the same animal cannot be treated the same statistically as eyes on different animals. Additionally a repeated measures ANOVA should have been used when looking at inter-group IOP as it appears that these are the same animals. That distinction is challenging as the orders state that they "divided the animals" when it appears as if they divided the data.The readability of the paper is a challenge and needs some support with English flow. given the amount of edits that are needed it is hard to go in depth. It is not clear why all of the raw data is included, animal body condition should be added and all of this raw data should be moved to the supplementary resources/materials. There is some confusing use of capital letters and formatting, both within the manuscript and throughout the reference section. The figures are missing numbers and appropriate in text referencing. You can't have a P value less 0 (line 211). In addition to language flow, the methods need to be outlined and results reanalyzed. Given those significant changes it feels inappropriate to give all the suggested edits line by line. If the authors can resolve the many issues I am happy to review the paper again and delve deeper. The authors should also remember to define acronyms the 1st time they are used.
Diabetes cases were discussed specifically these should be contrasted with data from the non-diabetes patients.
Comments on the Quality of English LanguageLine 45-48 are a great example of the language quality issues through out the paper. The information is there, but the word choice and flow make it challenging to read. For example events that should be in past tense are in present tense. I would recommend an editing service to clean up the flow and readability of the paper. These issues are compounded by inconsistent use of capitals/referencing and the inconsistent spacing between numbers and units. Overall copy editing and proofing are needed before this should be sent to reviewers again.
Author Response
Thank you very much for your review. The authors have made the necessary corrections and have endeavoured to do so.

Round 2
Reviewer 1 Report
Comments and Suggestions for Authors
The authors have satisfactorily adapted the manuscript to the suggested revisions.
Author Response
Thank you.
Reviewer 3 Report
Comments and Suggestions for Authors
The authors only responded to my comments about grammar and formatting and not those about experimental design, statistics and manuscript organization. The manuscript is not publishable until these are suitably addressed. Not addressing reviewer comments results in multiple review cycles, extended time to publication, and is taxing on both the editors and reviewers. Also some of the additions that the authors made included acronyms, that were not defined. Defining acronyms/abbreviations was something that was noted previously. Please make sure the next version addresses all issues and does not create new ones.
Comments on the Quality of English LanguageSeveral flow and syntax issues still exist. Consider using and outside editor service to help with the language fluidity and manuscript readability.
Round 3
Reviewer 3 Report
Comments and Suggestions for Authors
I think there was an error the authors sent/reposted the response to round one,
Author Response
Dear Review.
I am so grateful for your review. I have diligently corrected and repeated it.
thank you and best regard.
Round 4
Reviewer 3 Report
Comments and Suggestions for Authors
I have several concerns about this manuscript and the study data presented. The number of study groups differ between sections of the paper, and there are some significant inconsistencies in how the statistics were performed and reported throughout the paper. A lot of the analysis explained was flawed. I would encourage the authors to work with a statistician who can help them manage the data appropriately. I have outlined specific comments about the statistics and data below.
The manuscript has several writing errors. Numerous sections repeat information and/or phrases. Those are more than small grammar issues. There is also an abundance of small English errors. It is not the scientific reviewer’s job to be an English copy editor. I listed sections to be considered and specific lines with issues below. The authors should use a English editor before resubmitting any additional manuscript.
If the authors choose to resubmit. I request that they provide a written response for each of the following comments explaining how they have addressed the following issues.
Specific Comments (general):
Abstract suggests that the study animals were grouped into 2 groups but lines 263-270 suggest there were 3 groups. Please revise fix whichever sections are inaccurate and make changes to reflect that throughout the paper. Please include the sample size and list which animals/eyes were included in each group.
The article was changes into a study article not a “case presentation”. Please revise accordingly and remove phrase “case presentation” from line 19 and 99. Also remove the phrase “case study” from line 100, 108, 115. Remove “case report” from line 418, 421, 424. Please revise all these sentences appropriately and reword sections suggesting this addresses a single case.
Include animal medication and differences in medication prior to, during and after surgery. Any deviation in these protocols should be explored statistically, controlled for experimentally, discussed scientifically or excluded.
The study aims listed in 296-305 are different than the study aim listed in the introduction. These should be consistent and in line with the data reported. There are also many more aims listed than the study actually addressed.
Include what tests were performed prior to surgery (line 118).
How many patients received Mydrin-P, mannitol, paracentesis, and secondary surgery when was this applied? Were these animals excluded for subsequent readings?
There should be a clear methods section that include the statistics performed. Please include this prior to the results and discussion.
Specific Comments (statistics and data):
Table 1,2 & 3 are missing table numbers and table description.
Include animal body condition score in table 1.
Table 1 and 2 should be combined.
Figure 1 is missing a figure number and description
It is unclear what is different between figure 1 and 2 other than the results.
Formatting between figures should be corrected to be consistent (scales, lines, bolding ect)
For Figures 1-3. Why does the x axis say median? Does the Y axis represent the IOP from each animal? How many measurements were performed at each time. For example, were 3 measurements performed at 1 hour on each eye and the mean or median value recorded, or only one measurement.
Figure 3 is confusing. Or one of a series of values from each animal? There are also significant typing/editing and/or English issues in the caption. I am not sure these relate to the figure since they define terms that are not in the figure and nonsense like “**p<0.Pre OP:”
All three figures appear to have the same data. This is redundant, nor is it clear why all three were included.
Figures indicate median IOP but data reported is mean IOP (line 284). Why? This should be clear?
Was the data normally distributed? Include how that was tested. An independent t-test is different than a paired t-test. Please clarify what was performed. Where is this data presented/included? Line 279 suggests an alternative ANOVA was run than what is listed in line 395. What tests were actually performed and where is that reported and on what values (means, medians, etc.)? What type of post-hoc test was performed? Please include that information and the results?
What statistics were performed to look at the patients with diabetes compared with the other patients? What was done to control for the effect of diabetes? How many eyes were included for these 9 animals?
Data should be divided out by cataract stage for statistics and analysis. Cataract stage is known to impact IOP. These groups should be analyzed and reported on separately.
Each eye should not be treated independently how were eyes from the same animal treated differently statistically?
All of the raw data is included, animal body condition should be added and all of this raw data should be moved to the supplementary resources/materials
Data or Citations needed for sentences ending in line 67
Data or Citations needed for sentences ending in line 372
Data or Citations needed for sentences ending in line 384
Data or Citations needed for sentences ending in line 385
Data or Citations needed for sentences ending in line 388
Data or Citations needed for sentences ending in line 391
Specific Comments (English grammar, type errors and text duplications):
Remove all caps from 243-244. Why are capital letters used inconsistently (ex. Germany in line 137, but not South Korea line 144)
Repeat earlier sentence/information in lines 70-73
Repeat earlier sentence/information in lines 110-111
Repeat earlier sentence/information in lines 227-229
Repeat earlier sentence/information in lines 308-310
Do you mean glaucoma surgery in line 245?
Cut or revise sentence in line 140-141. Patients were not normal.
What is the relevance of items in line 74-77 consider cutting.
Revise run on sentences in lines 52 – 64
Revise run on sentences in lines 92-95
Revise run on sentences in lines 296-299
Revise English line 31
Revise English lines 67-72
Revise English lines 100-102
Revise English line 103 (#71?)
Revise English line 105, 308-310
Revise English line 311
Revise English lines 251-255
Revise English lines 260-261
Revise English lines 284-288
Revise English lines 307-310
Revise English lines 375-377
Author Response
Dear reviewer
Thank you very much for your review.There were so many reviews and frankly, it was very overwhelming. The administration of medication before, during, and after surgery was consistent in my hospital, and the English sentences were proofread by a professional.
All the reviews that have come in more than the first are difficult to proofread.
I have made the corrections to the best of my ability and would like to finalize the submission.
thank you and best regards.

Round 5
Reviewer 3 Report
Comments and Suggestions for Authors
Once again the authors have not addressed all of the reviewer comment, and they continue to not address those related to the reporting of data and statistics used in this study. That is critical to my approval (more than the writing quality). Until these are addressed I cannot recommend publications. Here is an example (see full list in my last review):
Figures indicate median IOP but data reported is mean IOP (line 284). Why? This should be clear?
Was the data normally distributed? Include how that was tested. An independent t-test is different than a paired t-test. Please clarify what was performed. Where is this data presented/included? Line 279 suggests an alternative ANOVA was run than what is listed in line 395. What tests were actually performed and where is that reported and on what values (means, medians, etc.)? What type of post-hoc test was performed? Please include that information and the results?
What statistics were performed to look at the patients with diabetes compared with the other patients? What was done to control for the effect of diabetes? How many eyes were included for these 9 animals?
Data should be divided out by cataract stage for statistics and analysis. Cataract stage is known to impact IOP. These groups should be analyzed and reported on separately.
Each eye should not be treated independently how were eyes from the same animal treated differently statistically?
All of the raw data is included, animal body condition should be added and all of this raw data should be moved to the supplementary resources/materials
Comments on the Quality of English LanguageThis has improved somewhat but there continues to be several issues many of those mentioned before were not addressed. Most of the issues involve tenses, syntax, run on sentences and repeated statement . Please have the paper edited before resubmission.
Author Response
Figures indicate median IOP but data reported is mean IOP (line 284). Why? This should be clear?
Paired tests were conducted to compare pre-surgery IOP with each post-surgery IOP measurement. The median IOP values for pre-surgery and post-surgery time points (1 hour to 8 weeks) were as follows: 15.23±11.32 pre-surgery, 17.45±12.42, 15.61±7.71, 11.81±9.03, 8.48±7.01, 11.39±12.39, 10.12±8.73, 9.58±7.90, 9.48±8.15, and 9.45±8.02.
Was the data normally distributed? Include how that was tested. An independent t-test is different than a paired t-test. Please clarify what was performed. Where is this data presented/included? Line 279 suggests an alternative ANOVA was run than what is listed in line 395. What tests were actually performed and where is that reported and on what values (means, medians, etc.)? What type of post-hoc test was performed? Please include that information and the results?
This study performed statistical tests randomly due to intra-case correlation and pre-sented the statistical values. For each test, P < 0.05 was considered statistically significant. After cataract surgery by PHACO, Statistical analysis was performed using a paired t-test to evaluate the changes over time from Pre-OP to Post-4W following the surgery. Addi-tionally, intraocular pressure (IOP) was compared among three groups: pre-surgery (Group A), 1 to 3 hours post-surgery (Group B), and 20 hours to 8 weeks post-surgery (Group C). The non-parametric Friedman test was used to evaluate overall differences be-tween the groups, followed by pairwise comparisons using the Wilcoxon signed-rank test.
Intraocular pressure (IOP) of the dogs that underwent surgery was measured according to the follow-up periods (Group A, B, and C). Values are presented as mean ± standard deviation (SD). P-values were obtained using the non-parametric Friedman test, followed by pairwise comparisons with the Wilcoxon signed-rank test. p < 0.05. Group A: preoperative data; Group B: data from 1 to 3 hours post-surgery; Group C: data from 20 hours to 8 weeks post-surgery.
What statistics were performed to look at the patients with diabetes compared with the other patients? What was done to control for the effect of diabetes? How many eyes were included for these 9 animals?
Based on your advice, the analysis was performed using only one eye (OD) from each case due to intracase correlation, and OS was selected when missing values were identified.
Data should be divided out by cataract stage for statistics and analysis. Cataract stage is known to impact IOP. These groups should be analyzed and reported on separately.
Thank you for your advice. In our study, the stage of cataract in Table 2 was briefly recorded.
Each eye should not be treated independently how were eyes from the same animal treated differently statistically?
Based on your advice, the analysis was performed using only one eye (OD) from each case due to intracase correlation, and OS was selected when missing values were identified.
All of the raw data is included, animal body condition should be added and all of this raw data should be moved to the supplementary resources/materials